# Viro-Immunological, Clinical Outcomes and Costs of Switching to BIC/TAF/FTC in a Cohort of People Living with HIV: A 48-Week Prospective Analysis

**DOI:** 10.3390/biomedicines10081823

**Published:** 2022-07-28

**Authors:** Maria Mazzitelli, Mattia Trunfio, Cristina Putaggio, Lolita Sasset, Davide Leoni, Sara Lo Menzo, Daniele Mengato, Anna Maria Cattelan

**Affiliations:** 1Infectious and Tropical Diseases Unit, Padua University Hospital, 35128 Padua, Italy; cristina.putaggio@aopd.veneto.it (C.P.); lolita.sasset@aopd.veneto.it (L.S.); davide.leoni@aopd.veneto.it (D.L.); sara.lomenzo@aopd.veneto.it (S.L.M.); annamaria.cattelan@aopd.veneto.it (A.M.C.); 2Infectious Disease Unit, Department of Medical Sciences at Amedeo di Savoia Hospital, University of Torino, 10124 Torino, Italy; mattia.trunfio@edu.unito.it; 3Hospital Pharmacy Department, Padua University Hospital, 35128 Padua, Italy; daniele.mengato@aopd.veneto.it

**Keywords:** keyword antiretroviral therapy, switch, naïve, bictegravir, BIC/FTC/TAF, real life, PLWH, elderly, discontinuation

## Abstract

To date, therapeutic switches are performed to reduce and prevent toxicity, improve adherence, promote virological control, and save costs. Drug switches are a daily challenge in the management of people living with HIV (PLWH), especially in those with multiple comorbidities and on polypharmacy. The objectives of this prospective analysis were: (I) to evaluate the viro-immunological efficacy of BIC/FTC/TAF in a cohort of PLWH who switched to this regimen from any other previous, at the Infectious and Tropical Diseases Unit of the Padua University Hospital; (II) to assess the impact on body weight, lipids, and renal function parameters at week 48; and (III) to evaluate daily costs changes, adherence, and the rate and causes of discontinuation of the regimen. We included all adult PLWH who switched to BIC/FTC/TAF from 1 February 2020 to 31 October 2021. We collected demographic, clinical, and laboratory data at baseline and week 48 after the switch. In addition, the estimated cART-related cost changes over the follow-up period were calculated. Over the study period, 290 individuals who switched to BIC/FTC/TAF, 76.9% were males, with a median age of 52 years, and 94.8% had an undetectable baseline HIV viremia. After a median time of 35 days (IQR: 1–55), 41 (14.1%) individuals discontinued the regimen. Factors significantly associated with discontinuation were switching from dual regimens, and neurological disorders. At week 48, we detected a significant increase in body weight, BMI, CD4 T-cell count, and CD4/CD8 ratio, and a significant reduction in triglycerides and costs; all patients had undetectable HIV RNA. Our results showed that switching to BIC/FTC/TAF may favor slightly immunological recovery and cost saving (−4.2 EUR/day from baseline to week 48, equivalent to a mean saving of 1533 EUR/year/person). The reduction in triglycerides does not appear to be clinically relevant, even if statistically significant, nor do both the increase in body weight and BMI (+1 kg and +0.29 BMI, respectively) and the increase in CD4 T-cell count (+45 cells/mmc). Further studies are needed to confirm our results.

## 1. Introduction

The introduction of combination antiretroviral therapy (cART) revolutionized the natural history of HIV disease, allowing people living with HIV (PLWH) to reach a life expectancy overlapping that of the general population [1,2]. Over the decades, antiretroviral regimens have undergone profound changes and innovations that made them more tolerated and adaptable to individuals’ needs [3]. In addition, newer antiretrovirals are much more effective in terms of virological control than those available in the past [3]. The pill burden has significantly been reduced over time, up to single-tablet regimens (STR), which favor adherence and retention in care [4,5]. Since cART is still a long-life therapy, therapeutic switches to reduce and prevent toxicity, improving adherence, and promoting virological control are the daily challenge in this new era. One of the most important issues for PLWH today is indeed the management and treatment of non-infectious comorbidities, as well as the management of drug–drug interactions (DDIs) [6,7,8]. In this regard, the class more associated with the perpetration of significant drug interactions was that of protease inhibitors (PIs), mainly due to the presence of the boosting agent, such as ritonavir and cobicistat [9]. The introduction of the class of integrase inhibitors (INSTIs), thanks to a higher genetic barrier and high potency in the absence of boosting agents, has allowed some limits that the PIs imposed to be overcome [10,11].

On the other hand, the use of this class remains preferable in subjects with multiple resistance mutations to classes of nucleoside inhibitors (NRTIs) and non-nucleoside (NNRTI) of reverse transcriptase [12]. INSTIs, in combination with NRTIs, entered, in combination as an STR, among the recommended regimens for the initiation of antiviral treatment in naïve individuals. In particular, the bictegravir/emtricitabine/tenofovir alafenamide combination (BIC/FTC/TAF) is recommended by all international guidelines for the treatment of HIV-1 infections in adults, adolescents, and children over 2 years of age and with a body weight ≥ 14 kg [13]. Data from clinical trials and from real life demonstrated that this combination is safe, well-tolerated, and able to provide virological control [14,15,16,17].

The main objective of this prospective study was to evaluate the viro-immunological and clinical efficacy of the BIC/FTC/TAF switching regimen in a cohort of individuals with HIV-1 infection attending the Infectious and Tropical Diseases Unit of the Padua University Hospital (from baseline to week 48). Secondary objectives were: (i) to assess the impact on body weight, lipids (total cholesterol, HDL, LDL, and triglycerides), and renal function parameters (from baseline to week 48); and (ii) to evaluate the level of adherence, the rate and causes of the discontinuation of this regimen, and the costs.

## 2. Materials and Methods

In this mono-centric prospective observational cohort study, we included all consecutive PLWH older than 18 years of age who switched to BIC/FTC/TAF combination from any previous dual or 2NRTIs-backbone-based antiretroviral regimen already ongoing for at least 6 months and regardless of plasma viremia. Individuals’ recruitment started on 1 February 2020, and ended on 31 October 2021, at the Infectious and Tropical Diseases Unit of Padua University Hospital (Italy).

We collected information on demographic (i.e., age, sex, and ethnicity), and clinical and HIV-related parameters (HIV acquisition risk, date of HIV diagnosis, previous AIDS events, and previous cART regimens). As per the study objective, the following information was collected at baseline (before BIC/FTC/TAF initiation) and 48 weeks after the switch: plasma HIV-RNA (virologic suppression or undetectability was defined as a limit of HIV RNA detection: <20 copies/mL), CD4+ T-cell count, CD4/CD8 ratio, creatinine, estimated glomerular filtration rate (eGFR), glucose, total cholesterol, triglycerides, HDL, LDL, body weight (kg), height (cm), and body mass index (BMI). Individuals were categorized by BMI class according to WHO definitions [18]. Undetectability was defined as an HIV-RNA < 50 copies/mL. Virological failure was defined as the presence of two consecutive viral loads higher than >50 copies/mL [19]. At failure, patients underwent an HIV resistance test by Sanger sequencing methods performed on plasma samples. The number of comedications and the presence of the following baseline comorbidities were recorded: hypertension, dyslipidemia, diabetes, osteoporosis, hepatitis co-infections, psychiatric and neurologic disorders, chronic renal failure, chronic obstructive pulmonary disease (COPD), and ischemic heart diseases. Multimorbidity was defined as the presence of two or more non-infectious comorbidities in the same patient and polypharmacy was defined as an intake of 5 or more non-HIV comedications per patient [20,21,22]. The level of adherence was assessed by using a self-administered questionnaire at baseline and during routine visits until week 48. For the subjects who discontinued the study regimen, the reason and time of discontinuation were recorded; individuals who were lost to follow-up before or at week 48 were considered discontinuations. Dietary pattern was asked at baseline and week 48 and categorized as: no food restrictions vs. specific diets (vegans, vegetarians, low carbs/fats diet prescribed by dieticians or doctors, ketogenic diet, and others). Physical exercise was also asked during baseline and week 48 evaluation and classified as: no activity, occasional physical exercise, or regular physical exercise (accordingly to WHO definition) [23].

Continuous and categorical variables were reported as median value (interquartile range) and absolute number (proportion), respectively. The intra-subject median changes from baseline to week 48 were analyzed by paired Wilcoxon test after excluding subjects classified as discontinuations; significance for changes in binary and multinomial variables was analyzed by the symmetry McNemar chi-square test and by the marginal homogeneity test, respectively. Inter-group comparisons were performed by Kruskal–Wallis test and chi-square test for trend. Binary and multinomial logistic regression was considered for univariate *p*-values < 0.1. The analyses were run through Stata SE 16 (Stata Corp LLC, College Station, TX, USA).

The study protocol received approval from the local Ethics Committee (n. 4052AO17) and written informed consent was obtained from each participant prior to enrollment in the study. In addition, the study was conducted accordingly to principles of good clinical practice and to the Helsinki declaration.

## 3. Results

### 3.1. Cohort Description

Over the study period, 290 individuals started the BIC/FTC/TAF. As reported in Table 1, 223 individuals were male (76.9%), while 67 individuals were female (33.1%), of which were 33 postmenopausal (49.2%). The median age was 52 years (44–58). A total of 236 (81.4%) persons were of Caucasian ethnicity, while the remaining were black-African in 45/290, Caribbean in 6/290, and Asian in 3/290 cases, respectively. The most common route of HIV acquisition was the sexual one, with transmission through homosexual and heterosexual intercourse in 57.6% and 24.8% of cases, respectively. The use of injecting drugs was present in 9.6% of cases. Median time since HIV diagnosis was 16 years (8–27). The median CD4+ T cell count at the nadir was 341 (178–536). About 12% of the individuals had a previous AIDS event in their past medical history. Most individuals (76.5%) were on INSTI-based regimens before switching. Most of the individuals (94.4%) were virologically suppressed with a good CD4 T-cell count (568 (IQR: 400–756) cell/mL). There were 54 individuals (18.6%) who presented with at least one NRTI mutation in their historical genotype resistance test (GRT), 48 (16.55%) had the M184V/I mutation, and 6 (2.1%) had the K65R mutation. Unfortunately, 43/290 (14.8%) of patients in our cohort did not have a genotypic resistance test before any cART initiation, since they were diagnosed with HIV from 1985 to 1996 (at that time genotyping assays were not available in our center). The most common reasons for switching to BIC/FTC/TAF were to perform a proactive switch (64.8% cases), to simplify the regimen into a single tablet (26.2%), and virological failure in 4.1% cases. Almost all (98.3%) individuals disclosed an adherence to antiretrovirals higher than 95%. The most frequent comorbidities in our cohort were dyslipidemia (45.2%), hypertension (22.8%), obesity (16.6%), osteoporosis (11.7%), and chronic kidney disease (11.4%). Polypharmacy was present in 8.6% of individuals. One hundred and sixteen individuals (40%) had multimorbidity. As for lifestyle, 85.5% were on a diet with no restriction, and 65.2% had a sedentary life.

### 3.2. Viroimmunological Profile Evolution

At week 48, the intention-to-treat (ITT) analysis showed that the BIC/FTC/TAF regimen warranted virologic suppression in 95% of individuals; while in the per-protocol analysis population, no patient had HIV-1 RNA > 20 copies/mL at the end of follow-up, including those subjects 14 (5.6%) who were on virological failure before the switch. No discontinuation of the regimen due to virological failure was observed during the study period.

The median CD4+ T cell count significantly increased from 568 (IQR: 400–756) cells/L at baseline to 610 (IQR: 450–780) cells/L after 48 weeks of follow-up. A statistically significant increase in CD4+/CD8+ ratio was also detected, from 0.72 (IQR: 0.48–1.16) to 0.81 (IQR: 0.56–1.19) (*p* < 0.001). A questionnaire on self-reported antiretroviral adherence did not show any differences from baseline to the follow-up time point (Table 2).

### 3.3. Rate of Discontinuation of BIC/FTC/TAF and Factors Associated with Discontinuation

Overall, 41 individuals (14.1%) discontinued the study regimen: 15 (5.2%) individuals for toxicity/intolerance, 6 (2.1%) subjects who were previously pro-actively simplified to a dual regimen containing dolutegravir and lamivudine, 1 (0.3%) subject for weight gain greater than 5% from the baseline, 2 (0.7%) subjects for both weight gain and intolerance, 2 (0.7%) women because of pregnancy, and 15 (5.2%) individuals who were lost to follow-up. In subjects who discontinued the regimens for toxicity, the median time from switch to discontinuation was 35 days (IQR:1–55). Adverse events causing discontinuation (detected in 17/41 individuals, 41.4%) were gastrointestinal disorders (9/17, 52.9%), sleep disturbances (4/17, 23.5%), neuropsychiatric disorders (3/17, 17.6%), elevation of liver enzymes (2/17, 11.7%), difficulty to swallow the tablet (1/17, 5.9%), itching (1/17, 5.9%), myalgia (1/17, 5.9%), and erectile disfunction (1/17, 5.9%), with four subjects experiencing more than one adverse event. When we compared individuals who maintained the regimen with those who continued it (Table 1), we detected a significant difference in HCV antibody positivity prevalence, length of HIV infection, neurological and psychiatric disorders prevalence, and polypharmacy; a mild trend was also observed for the type of regimen before switching to BIC/FTC/TAF (*p* = 0.057). Univariate and multivariate results are shown in Table 3 discontinuation was associated with being on a previous dual therapy (aOR 3.392 [1.266–9.090], *p* = 0.015 compared to switching from an INI-based regimen) and with the presence of neurological disorders, independently from age, sex, ethnicity, HCV serostatus, polypharmacy, and psychiatric disorders.

### 3.4. Changes from Baseline to Follow-Up of Body Weight, BMI, Metabolic Parameters, and Costs

Table 3 depicts the median changes from baseline to week 48 of the different clinical and laboratory parameters among the 249 PLWH who were still on BIC/FTC/TAF at week 48. No changes were detected for lipid profile (cholesterol, LDL, HDL, and triglycerides), kidney function, and glomerular filtration rate. Over the study period, we did not observe significant differences from baseline to follow-up among the BMI categories (Figure 1). However, 25/249 (16.7%) individuals experienced an increased BMI, 14 (5.6%) individuals experienced a decrease in BMI, while (210/249) 77.7% remained stable. The three groups of the type of BMI change at week 48 differed only for baseline BMI, both as a continuous parameter and as BMI classes distribution (Table 4). Nevertheless, baseline BMI, neither as a continuous variable nor as a categorical class, was found to be independently associated with BMI change at week 48, when multinomial regression models were adjusted for age, sex, ethnicity, diets, and physical activity at any combination (data not shown). Regarding treatment costs, if compared with the baseline regimen, switching to BIC/FTC/TAF allowed the saving of a mean of EUR 4.2 for each day of treatment, with an overall amount of saving over the study period (48 weeks) of 1533 EUR/patient/year). However, overall, we need to consider that 41 (14.1%) individuals discontinued the study regimen: 22/41 (53.6%) came back to the pre-switch regimen (median cost per day EUR 26.1, IQR 19.1–26.5), and 6 (14.6%) individuals were simplified to the dual regimen dolutegravir/lamivudine (24 EUR/day).

## 4. Discussion

BIC/FTC/TAF is one of the latest STRs introduced in clinical practice including a high genetic barrier integrase strand transfer inhibitor with TAF/FTC. It is recommended by most antiretroviral therapy guidelines both as an initiation and switch regimen in PLWH. Even though its virological efficacy was demonstrated in large double-blind randomized clinical trials, prospective real-life data assessing durability, impact on viro-immunological, clinical, metabolic parameters, and costs of BIC/FTC/TAF prescribed as switch regimen are still scarce. In our cohort in an analysis, the switch to BIC/FTC/TAF was associated with an overall maintenance of virologic suppression in 95% of individuals, and no virological failure in the 249 subjects who completed the 48 weeks of follow-up was observed. No patients discontinued the regimen due to VF during the time of observation, and viral suppression was maintained also in individuals with virological failure at baseline regardless of the pre-existing NRTI resistance such as M184V and K65R. These results further support the high resistance barrier of bictegravir as previously demonstrated in both in vitro and clinical studies [16,24,25]. In addition, BIC/FTC/TAF was successful in maintaining an excellent immunological efficacy as a switch regimen; at week 48, the median CD4 increased to 610 (450–780) cells/mL, and the CD4+/CD8+ ratio slightly improved, confirming the persistence and further mild amelioration of immune recovery after switching. Adverse events and lost to follow-up almost equally contributed to the discontinuation rate in our cohort of individuals. Recently, results from a Spanish cohort, with a shorter follow-up, and including also naïve individuals, showed that at the end of the study, 88% of subjects remained on the study regimens; 42/1584 (2.6%) discontinued treatment for toxicities, and 7/1584 (0.4%) experienced virological failure [26].

In clinical trials performed on naïve individuals, which reached 5 years of follow-up, the rate of discontinuation was around 11–13% [27], occurring in 34 and 28 subjects of Study 1489 and 1490, respectively. In this study, most discontinuation occurred due to participants’ decisions and for loss to follow-up, while toxicities were reported in a small percentage of individuals (4/252, 1.6%) who discontinued the regimen in study 1489. In our cohort, discontinuations due to toxicities were recorded in 17/290 (5.8%) individuals, a proportion much higher than that reported by clinical trials and by other smaller cohort studies. They were mainly due to gastrointestinal side effects (52.9%) and sleep disturbances/neuropsychiatric disorders (23.5% and 17.6%).

In the multivariate analysis, neurological disorders were significantly associated with drug discontinuation. Indeed, it is well known that central nervous system (CNS)-related adverse events have been associated with the use of INSTI-based treatment, including DTG, for which the incidence of insomnia and headache ranged from 3% to 15 [28,29,30]. Most cohort studies reported a higher rate of discontinuation of DTG due to neuropsychiatric adverse events in comparison with other INSTIs. However, a recent study showed that the short-term tolerability of BIC/FTC/TAF was comparable to DTG-containing regimens, with a low rate of cross-intolerance between the two INSTIs [31].

Of note, the long HIV history of our patients associated with the large variety of previous antiretroviral agents precluded a meaningful description of the role bictegravir may have had on CNS-related AEs. Interestingly, none of the patients who withdrew BIC/FTC/TAF for CNS-related AEs were older than 50 years old, confirming good tolerability of the regimen in aging people (data not shown).

We believe that in our study the higher proportion of participants who withdrew because of adverse events may be somewhat related to the COVID-pandemic period during which the study was conducted. It is known how COVID-19 had an important impact on the “HIV world” (involving prevention, diagnosis, and treatment), and it is still also debated whether people with HIV have a poorer prognosis than the general population [32,33,34]. The modified lifestyle due to the lockdown curfew, the fear of self-isolation with the feelings of more stress and anxiety may have had a negative impact on the patient’s psychosocial health and may have triggered an increased perception of severe side effects related to the new drug regimen.

Among individuals who were considered as discontinuation, the loss to follow-up rate (15/290, 5.1%) was also very similar to the one reported by clinical trials in naïve individuals. However, in our cohort, most lost to follow-up individuals were black-African people (migrant status) who represented about 20% of the study population. This finding is not surprising considering that in our region black people are often migrants, and a difficult-to-treat population, with low levels of adherence and high rates of loss to follow-up. Furthermore, we believe that these results draw us closer to the unmet needs of this subgroup of people to which we should likely guarantee a better linkage to care. In this specific population, the efficacy, tolerability, and lack of resistance to BIC/FTC/TAF may represent a good treatment option.

Switching to BIC/FTC/TAF did not result in a significant reduction in total cholesterol, LDL cholesterol, and triglycerides in our cohort (Table 2). In previous clinical studies, switching to BIC/FTC/TAF has been associated with variable changes in lipid parameters, showing a significant decline in triglycerides and total cholesterol to HDL ratio through week 48 when switching from regimens containing boosted PIs, but not when switching from regimens containing DTG/ABC/3TC [35,36].

In our study most individuals at baseline were treated with INSTI and TAF; therefore, it is reasonable that switching to BIC/FTC/TAF had a “neutral” lipidic effect, even if in the absence of the known lipid-lowering effects of TDF compared with TAF and in the absence of lifestyle modifications.

In fact, in our study, we explored for the first time, the changes/lifestyle modifications (i.e., diet and exercise) from baseline to follow-up.

Indeed, the failure to change the lifestyle (introduction of a specific diet and improving physical exercise) may have a role in the significant increase in BMI in our cohort of individuals. At month 12 of follow-up, a mean increase of +1 kg was detected and 17% of individuals showed a significant increase in the BMI value. Even though the weight gain observed in our study overlaps that foreseen by the WHO in the general population [37], these results may raise some concerns. First, most individuals who switched to the study regimens were previously on TAF/FTC and integrase inhibitors, so it is likely that we could have missed with our observation a significant weight gain that had already occurred in the past. In the Swiss cohort, it has been reported that the replacement of TDF with TAF was associated with weight increase and the development of obesity [38]. Specifically, among individuals with a normal BMI (51.7% of the entire cohort), 13.8% who switched to TAF became overweight/obese, compared with 8.4% of those continuing TDF [38].

Secondly, it is likely that the study regimen does not imply a clinically relevant weight gain over 12 months. It is interesting to note that in our population about 38% of individuals were overweight, with 12% obese at study enrollment. These data are shared with other large HIV cohorts, such as the “Opera” cohort where among a total of 6908 virological suppressed PLWH, overweight (BMI ≥ 25 to <30) and obesity (BMI > 30) were reported in 37–39% and 26–29% of participants, respectively [36]. It is well known that median BMI and the prevalence of obesity among PLWH initiating ART have been steadily increasing and these trends are attributable to several factors, including the “return to health” weight gain following ART initiation, the recent INSTI-based regimen, in addition to the dietary habits and lifestyle among people with HIV. In fact, it has been demonstrated that a large proportion of PLWH does not meet the current diet and physical activity recommendations that could help in the prevention and mitigation of metabolic comorbidities in this population [39,40,41,42,43].

It is difficult to speculate on a possible evolution of all the different parameters over time. However, it is likely that patients may put further weight on during a longer follow-up; from of a mean 1 kg/year as already shown in the general population, to 5–6 kg during 5 years of follow-up as already observed in the above-mentioned clinical trials on INSTIs combined with tenofovir alafenamide.

However, further studies are needed to evaluate the mechanism of weight gain among individuals starting INSTI-based regimens. In the meantime, clinicians have to be aware that the development of overweight and obesity in the HIV population may significantly increase the risk of cardiovascular events and should be prevented. In this context, the robustness and the convenience of the BIC/TAF/FTC regimen may be integrated into care interventions and counseling oriented towards a healthy lifestyle.

Finally, given the economic impact of costly brand antiretroviral treatments in high-income countries, the BIC/TAF/FTC regimen in our cohort of individuals allowed an overall amount of saving of 1533 EUR/patient/year; it was not a budget-impact analysis but only/simply a direct analysis of the daily cost of the regimen. However, we obtained the result of a successful regimen without de-simplifying a single tablet antiretroviral combination, often considered a cost-saving strategy, and without compromising the best quality of care. This aspect needs to be considered in terms of resource-saving, allowing clinicians to consider not only the medical concerns but also cost-saving strategies, warrantying safe switches at a lower cost.

Unfortunately, our study is somewhat limited by the low numbers, and by the high rates of individuals who were lost to follow-up. Moreover, we assessed only the raw cost of switching, without performing a proper cost–benefit analysis. In conclusion, in our study, BIC/FTC/TAF has been demonstrated to be an excellent option for switching therapy in HIV people presenting in different clinical contexts and in individuals presenting with metabolic disorders.

## Figures and Tables

**Figure 1 biomedicines-10-01823-f001:**
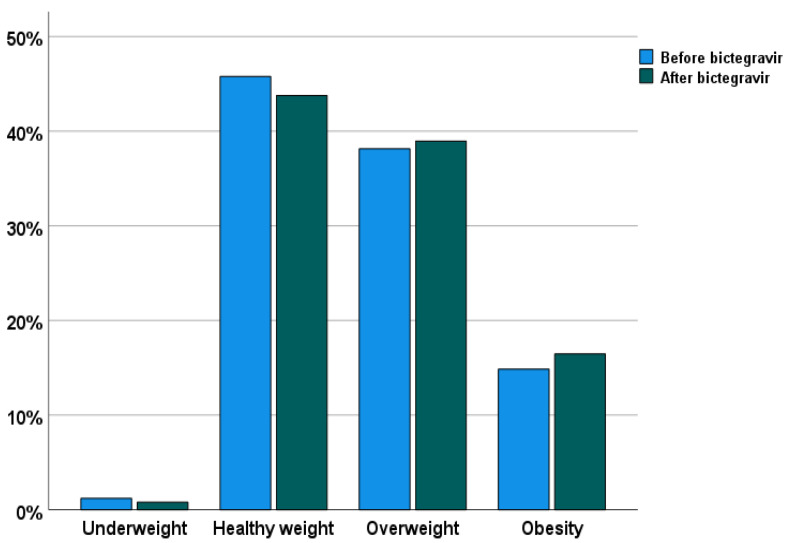
Proportion of individuals according to BMI categories before and after switching to BIC/FTC/TAF. No statistically significant differences were detected among BMI categories.

**Table 1 biomedicines-10-01823-t001:** Comparison of baseline characteristics between subjects who discontinued BIC/FTC/TAF during the study period and subjects who remained on BIC/FTC/TAF at week 48.

Parameter	Study Population (*n* = 290)	BIC Discontinuation (*n* = 41)	BIC Ongoing (*n* = 249)	*p*
Age, years, median (IQR)	52 (44–58)	54 (48–62)	52 (44–57)	0.500
Caucasian, *n* (%)	236 (81.4)	36 (87.8)	200 (80.3)	0.255
Gender, male, *n* (%)	223 (76.9)	30 (73.2)	193 (77.5)	0.542
Risk factor, *n* (%)				0.417
MSM	167 (57.6)	25 (60.9)	142 (57.0)	
Heterosexual	72 (24.8)	11 (26.8)	61 (24.5)	
IVDU	28 (9.6)	5 (12.2)	23 (9.2)	
Others	23 (7.9)	0 (0.0)	23 (9.2)	
Length of HIV infection, years, median (IQR)	16 (8–27)	13 (8–24)	17 (8–28)	0.048
Previous AIDS episodes, yes, *n* (%)	35 (12.1)	5 (12.2)	30 (12.0)	0.979
CD4 nadir, cells/mmc, median (IQR)	341 (178–536)	421 (209–520)	320 (178–542)	0.236
CD4 T cells count, cells/mmc, median (IQR)	571 (396–759)	615 (379–830)	568 (400–756)	0.736
CD4/CD8 ratio, median (IQR)	0.71 (0.47–1.14)	0.70 (0.45–1.01)	0.72 (0.48–1.16)	0.651
HIV-RNA, cp/mL, median (IQR)	0 (0–0)	0 (0–0)	0 (0–0)	0.637
Detectable plasma HIV-RNA, *n* (%)	15 (5.2)	1 (2.4)	14 (5.6)	0.395
Regimen before switch, *n* (%)				0.057
Dual	25 (8.6)	8 (19.5)	17 (6.8)	
nNRTI-based	17 (5.9)	1 (2.4)	16 (6.4)	
PI-based	26 (8.9)	5 (12.2)	21 (8.4)	
INSTI-based	222 (76.5)	27 (65.8)	195 (78.3)	
From TDF	14 (4.8)	2 (4.8)	12 (4.8)	0.987
From TAF	220 (75.9)	28 (68.3)	192 (77.1)	0.222
From PIs	41 (14.1)	9 (21.9)	32 (12.8)	0.122
From COBI	180 (62.1)	26 (63.4)	154 (61.8)	0.848
From INSTIs	232 (80.0)	30 (73.2)	202 (81.1)	0.239
Self-reported optimal adherence, *n* (%)	285 (98.3)	39 (95.1)	246 (98.8)	0.797
Resistance mutations, *n* (%)				0.442
PI	7 (2.4)	2 (4.9)	5 (2.0)	0.267
INSTIs	1 (0.3)	0 (0.0)	1 (0.4)	0.681
NNRTI	25 (8.6)	6 (14.6)	19 (7.6)	0.138
NRTI	49 (16.9)	6 (14.6)	43 (17.3)	0.675
M184V/I	6 (2.1)	0 (0.0)	6 (2.4)	0.315
Comorbidity/subject, median (IQR)	1 (0–2)	2 (1–2)	1 (0–2)	0.158
Comorbidities, *n*, (%)				
Hypertension	66 (22.8)	8 (19.5)	58 (23.3)	0.593
Dyslipidemia	131 (45.2)	20 (48.8)	111 (44.6)	0.617
Ischemic heart disease	14 (4.8)	2 (4.8)	12 (4.8)	0.987
Chronic renal failure	33 (11.4)	8 (19.5)	25 (10.0)	0.077
Diabetes mellitus	19 (6.6)	3 (7.3)	16 (6.4)	0.831
CNS disorders	18 (6.2)	8 (19.5)	10 (4.0)	<0.001
COPD	14 (4.8)	4 (9.8)	10 (4.0)	0.113
Osteoporosis	34 (11.7)	2 (4.8)	32 (12.8)	0.142
Psychiatric disorders	30 (10.3)	10 (24.4)	20 (8.0)	0.002
Positive HCV Ab, *n*, (%)	54 (18.6)	13 (31.7)	41 (16.5)	0.020
HBsAg, *n,* (%)	48 (16.5)	9 (21.9)	39 (15.7)	0.316
Multimorbidity, *n* (%)	116 (40.0)	21 (51.2)	95 (38.1)	0.114
Polypharmacy, *n* (%)	25 (8.6)	7 (17.1)	18 (7.2)	0.038
Weight, kg, median (IQR)	76.0 (70.0–82.0)	75.0 (69.5–81.0)	76.0 (70.5–82.0)	0.889
BMI, median (IQR)	25.3 (23.3–27.7)	25.4 (23.7–26.5)	25.2 (23.2–27.8)	0.924
BMI class, *n* (%)				0.511
<18.5 kg	4 (1.4)	1 (2.4)	3 (1.2)	
18.5–24.9 kg	131 (45.2)	17 (41.5)	114 (45.8)	
25.0–29.9 kg	118 (40.7)	23 (56.1)	95 (38.1)	
≥30.0 kg	37 (12.7)	0 (0.0)	37 (14.8)	
Diet, n (%)				0.994
No food restrictions	248 (85.5)	35 (85.)	213 (85.5)	
Vegan/vegetarian	9 (3.1)	1 (2.4)	8 (3.2)	
Ketogenic	15 (5.2)	2 (4.8)	13 (5.2)	
Low carbs/fats	11 (3.8)	2 (4.8)	9 (3.6)	
Others	7 (2.4)	1 (2.4)	6 (2.4)	
Physical activity, *n* (%)				0.334
None	189 (65.2)	29 (70.1)	160 (64.2)	
Occasional	13 (4.5)	3 (7.3)	10 (4.0)	
Regular	88 (30.3)	9 (21.9)	79 (31.7)	
Total cholesterol, mmol/L, median (IQR)	4.75 (4.09–5.51)	4.70 (4.09–5.58)	4.76 (4.09–5.51)	0.999
LDL, mmol/L, median (IQR)	3.08 (2.45–3.66)	3.09 (2.45–3.78)	3.08 (2.45–3.62)	0.921
HDL, mmol/L, median (IQR)	1.27 (1.03–1.51)	1.33 (1.03–1.61)	1.26 (1.02–1.47)	0.211
Triglycerides, mmol/L, median (IQR)	1.28 (0.88–1.95)	1.33 (0.96–2.08)	1.28 (0.86–1.92)	0.963
Serum creatinine, mmol/L, median (IQR)	86.0 (74.8–100.0)	86.0 (68.5–104.0)	86.0 (75.0–99.5)	0.921
eGFR, mL/min, median (IQR)	87.0 (73.0–99.0)	91.0 (69.0–102.5)	87.0 (73.5–99.0)	0.657
Reasons for switch, *n* (%)				0.742
Proactive	188 (64.8)	26 (63.4)	162 (65.1)	
Switch to STR	76 (26.2)	10 (24.4)	66 (26.5)	
DDIs	9 (3.1)	2 (4.8)	7 (2.8)	
Viral failure	12 (4.1)	1 (2.4)	11 (4.4)	
Drug toxicity	5 (1.7)	2 (4.8)	3 (1.2)	

MSM = Men who have sex with men, IVDU = intravenous drug use, IQR = interquartile range, nNRTI = non-nucleos(t)ide reverse transcriptase inhibitors, PI = protease inhibitors, INSTIs = integrase inhibitors, STR = single tablet regimen, DDI = drug–drug interaction, BMI = body mass index, eGFR=estimated glomerular filtration rate, COBI= cobicistat, COPD= chronic obstructive pulmonary disease. Non-parametric tests were applied according to data distribution and type of variables: Mann–Whitney U test (for continuous variables) and Chi-square test (for categorical variables).

**Table 2 biomedicines-10-01823-t002:** Median difference in viro-immunological and metabolic parameters between baseline and W48 of the study cohort.

Parameter	Baseline *n* = 249	W48 *n* = 249	Median Difference (95%CI)	*p*
CD4 T cells count, cells/mmc, median (IQR)	568 (400–756)	610 (450–780)	+45 (27; 61)	<0.001
CD4/CD8 ratio, median (IQR)	0.72 (0.48–1.16)	0.81 (0.56–1.19)	+0.065 (0.050; 0.085)	<0.001
HIV-RNA, cp/mL, median (IQR)	0 (0–0)	0 (0–0)	0.0 (0; 0)	0.522
Detectable plasma HIV-RNA, *n* (%)	14 (5.6%)	0 (0.0%)	-	<0.001
Self-reported optimal adherence, *n* (%)	246 (98.8%)	246 (98.8%)	-	0.989
Cost (EUR/day), median (IQR)	26.6 (20.0–26.6)	20.0 (20.0–20.0)	−4.2 (−5.2; −3.3)	<0.0005
Weight (Kg), median (IQR)	76.0 (70.5–82.0)	77.0 (71.0–84.0)	+1.0 (0.5; 1.2)	<0.001
BMI, median (IQR)	25.2 (23.2–27.8)	25.4 (23.6–27.8)	+0.29 (0.15; 0.40)	<0.001
BMI class, *n* (%)			-	0.078
<18.5 kg	3 (1.2)	2 (0.8)		
18.5–24.9 kg	114 (45.8)	109 (43.8)		
25.0–29.9 kg	95 (38.1)	97 (38.9)		
≥30.0 kg	37 (14.8)	41 (16.5)		
Diet, *n* (%)			-	0.959
No food restrictions	213 (85.5)	208 (83.5)		
Vegan/vegetarian	8 (3.2)	8 (3.2)		
Ketogenic	13 (5.2)	15 (6.0)		
Low carbs/fats	9 (3.6)	12 (4.8)		
Others	6 (2.4)	6 (2.4)		
Physical activity, *n* (%)			-	0.478
None	160 (64.2)	150 (60.2)		
Occasional	10 (4.0)	15 (6.0)		
Regular	79 (31.7)	84 (33.7)		
Total cholesterol, mmol/L, median (IQR)	4.76 (4.09–5.51)	4.70 (4.15–5.48)	+0.020 (−0.070;0.11)	0.690
LDL, mmol/L, median (IQR), mmol/L, median (IQR)	3.08 (2.45–3.62)	3.01 (2.44–3.66)	−0.015 (−0.085; 0.055)	0.667
HDL	1.26 (1.02–1.47)	1.24 (1.05–1.52)	+0.005 (−0.020; 0.035)	0.665
Triglycerides, mmol/L, median (IQR)	1.28 (0.86–1.92)	1.19 (0.87–1.70)	−0.12 (−0.20; −0.035)	0.003
Serum creatinine, mmol/L, median (IQR)	86.0 (75.0–99.5)	86.0 (75.0–99.5)	−0.50 (−1.5; 1.0)	0.624
eGFR, mL/min, median (IQR)	87.0 (73.5–99.0)	87.0 (72.0–97.0)	−0.50 (−1.5; 1.0)	0.681

IQR = interquartile range, eGRF = estimated glomerular filtration rate.

**Table 3 biomedicines-10-01823-t003:** Univariate and Multivariate analysis for BIC/FTC/TAF discontinuation.

	Univariate Analysis		Multivariate Analysis	
	OR (95%CI)	*p*	aOR (95%CI)	*p*
Age, per year more	1.024 (0.995–1.055)	0.499	1.010 (0.977–1.045)	0.549
Ethnicity, ref. Caucasian	0.567 (0.211–1.520)	0.295	0.630 (0.206–1.928)	0.418
Gender, ref. woman	1.264 (0.596–2.681)	0.542	1.293 (0.535–3.123)	0.569
Risk factor,			-	-
MSM	Ref	-		
Heterosexual	1.024 (0.474–2.212)	0.951		
IVDU	1.235 (0.429–3.551)	0.696		
Others	0.0 (0.0–0.0)	0.998		
Length of HIV infection, per year more	0.978 (0.947–1.010)	0.173	-	-
Previous AIDS episodes, ref. none	1.014 (0.369–2.784)	0.979	-	-
CD4 nadir, cells/mmc, per unit more	1.000 (0.998–1.002)	0.238	-	-
CD4 T cells count, cells/mmc, per unit more	1.000 (0.998–1.001)	0.736	-	-
CD4/CD8 ratio, per unit more	0678 (0.333–1.379)	0.651	-	-
HIV-RNA, cp/mL, per unit more	0.999 (0.998–1.001)	0.635	-	-
Plasma HIV-RNA, ref. undetectable	0.420 (0.054–3.280)	0.408	-	-
Regimen before switch,				
Dual	Ref	-	Ref	-
nNRTI-based	0.133 (0.015–1.184)	0.071	0.130 (0.013–1.259)	0.078
PI-based	0.506 (0.140–1.833)	0.300	0.495 (0.123–1.991)	0.322
INSTI-based	0.294 (0.116–0.747)	0.010	0.295 (0.110–0.790)	0.015
From TDF, ref. no	1.013 (0.218–4.700)	0.987	-	-
From TAF, ref. no	0.639 (0.311–1.315)	0.224	-	-
From PIs, ref. no	1.907 (0.834–4.363)	0.126	-	-
From COBI, ref. no	1.069 (0.539–2.121)	0.848	-	-
From INSTIs, ref. no	0.635 (0.297–1.357)	0.241	-	-
Self-reported optimal adherence, ref. yes	0.936 (0.420–1.894)	0.801	-	-
Comorbidity/subject, per unit more	1.179 (0.925–1.503)	0.156	-	-
Comorbidities,				
Hypertension, ref. no	0.798 (0.349–1.824)	0.593	-	-
Dyslipidemia, ref. no	1.184 (0.611–2.294)	0.617	-	-
Ischemic heart disease, ref. no	1.013 (0.218–4.700)	0.987	-	-
Chronic renal failure, ref. no	2.172 (0.905–5.216)	0.083	-	-
Diabetes mellitus, ref. no	1.150 (0.320–4.135)	0.831	-	-
CNS disorders, ref. no	5.794 (2.135–15.724)	<0.001	4.685 (1.493–14.705)	0.008
COPD, ref. no	2.584 (0.770–8.666)	0.114	-	-
Osteoporosis, ref. no	0.348 (0.080–1.510)	0.142	-	-
Psychiatric disorders, ref. no	3.661 (1.570–8.539)	0.003	1.777 (0.595–5.309)	0.303
HCV serology, ref. negative	2.355 (1.126–4.928)	0.023	1.624 (0.694–3.802)	0.263
HbsAg, ref. negative	1.514 (0.671–3.420)	0.318	-	-
Multimorbidity, ref. no	1.702 (0.877–3.305)	0.116	-	-
Polypharmacy, ref. no	2.642 (1.028–6.793)	0.044	1.166 (0.335–4.057)	0.809
Weight, Kg, per kg more	0.979 (0.951–1.009)	0.890	-	-
BMI, per unit more	0.945 (0.860–1.083)	0.924	-	-
BMI class,			-	-
<18.5 kg	Ref	-		
18.5–24.9 kg	0.447 (0.044–4.552)	0.497		
25.0–29.9 kg	0.726 (0.072–7.307)	0.786		
≥30.0 kg	0.000 (0.000–0.000)	0.998		
Diet,			-	-
No food restrictions	Ref	-		
Vegan/vegetarian	0.761 (0.092–6.271)	0.799		
Ketogenic	0.936 (0.203–4.328)	0.933		
Low carbs/fats	1.352 (0.280–6.522)	0.707		
Others	1.014 (0.119–8.681)	0.990		
Physical activity,			-	-
None	Ref	-		
Occasional	1.655 (0.429–6.381)	0.464		
Regular	0.629 (0.284–1.392)	0.252		
Total cholesterol, mmol/L, per unit more	0.998 (0.733–1.360)	0.992	-	-
LDL, mmol/L, per unit more	0.972 (0.671–1.405)	0.921	-	-
HDL, mmol/L, per unit more	1.134 (0.750–1.716)	0.211	-	-
Triglycerides, mmol/L, per unit more	1.109 (0.930–1.323)	0.964	-	-
Serum creatinine, mmol/L, per unit more	1.008 (0.993–1.022)	0.921	-	-
eGFR, mL/min, per unit more	1.002 (0.982–1.022)	0.660	-	-
Reasons for switch,			-	-
Proactive	Ref	-		
Switch to STR	0.944 (0.431–2.067)	0.885		
DDIs	4.154 (0.662–26.063)	0.129		
Viral failure	0.566 (0.070–4.573)	0.594		
Drug toxicity	1.780 (0.351–9.042)	0.487		

Binary logistic regression that included significant univariate variables plus age, sex, and ethnicity. MSM = Men who have sex with men, IVDU = intravenous drug use, IQR = interquartile range, NNRTI = non-nucleos(t)ide reverse transcriptase inhibitors, PI = protease inhibitors, INSTIs = integrase inhibitors, STR = single tablet regimen, DDIs = drug–drug interaction, BMI = body mass index.

**Table 4 biomedicines-10-01823-t004:** Comparison among the three individual categories identified based on the change in BMI class from baseline to W48.

Characteristic	Increased BMI Class (*n* = 25)	Decreased BMI Class (*n* = 14)	No Change in BMI class (*n* = 210)	*p*
Age, years, median (IQR)	53 (46–56)	50 (36–55)	52 (44–58)	0.719
Caucasian, *n* (%)	18 (72.0)	12 (85.7)	170 (80.9)	0.497
Male sex, *n*, (%)	17 (68.0)	12 (85.7)	155 (73.8)	0.392
Risk factor, *n* (%)				0.752
MSM	13 (52.0)	7 (50.0)	122 (58.1)	
Heterosexual	9 (36.0)	3 (21.4)	49 (23.3)	
IVDU	3 (12.0)	2 (14.3)	18 (8.6)	
Others	0 (0)	2 (14.3)	21 (10.0)	
Length of HIV infection, years, median (IQR)	21 (7–30)	13 (11–26)	17 (8–27)	0.169
CD4+T cell count at nadir, cells/mmc, median (IQR)	355 (168–561)	350 (161–569)	314 (183–538)	0.245
CD4+ T cells count, cells/mmc, median (IQR)	582 (342–741)	541 (347–709)	571 (401–778)	0.549
CD4/CD8 ratio, median (IQR)	0.70 (0.52–1.06)	0.61 (0.39–1.01)	0.74 (0.47–1.18)	0.491
HIV-RNA, copies/mL, median (IQR)	0 (0–0)	0 (0–0)	0 (0–0)	0.581
Detectable plasma HIV-RNA, *n* (%)	1 (4.0)	0 (0.0)	13 (6.1)	0.582
Previous AIDS episodes, *n* (%)	4 (16.0)	0 (0.0)	26 (12.4)	0.317
Regimen before switch, *n* (%)				
Dual	3 (12.0)	0 (0.0)	14 (6.7)	
nNRTI-based	1 (4.0)	0 (0.0)	15 (7.1)	0.131
PI-based	1 (4.0)	0 (0.0)	20 (9.5)	
INSTI-based	21 (84.0)	14 (100)	161 (76.7)	
From TDF, *n* (%)	1 (4.0)	0 (0.0)	11 (5.2)	0.663
From TAF, *n* (%)	17 (68.8)	13 (92.8)	162 (77.1)	0.209
From PIs, *n* (%)	5 (20.0)	0 (0.0)	27 (12.8)	0.203
From COBI, *n* (%)	13 (52.0)	12 (85.7)	129 (61.4)	0.111
From INSTIs, *n* (%)	20 (80.0)	14 (100)	168 (80.0)	0.179
Self-reported optimal adherence, *n* (%)	25 (100)	14 (100)	207 (98.6)	0.388
Comorbidity/subject, median (IQR)	1 (1–2)	1 (1–2)	1 (0–2)	0.760
Comorbidities, *n* (%)				
Hypertension	5 (20.0)	3 (21.4)	50 (23.8)	0.901
Dyslipidemia	9 (36.0)	6 (42.8)	96 (45.7)	0.648
Ischemic heart disease	3 (12.0)	0 (0.0)	9 (4.3)	0.162
Chronic renal failure	2 (8.0)	0 (0.0)	23 (10.9)	0.228
Diabetes mellitus	1 (4.0)	0 (0.0)	15 (7.1)	0.502
CNS disorders	2 (8.0)	0 (0.0)	8 (3.8)	0.442
COPD	1 (4.0)	1 (7.1)	8 (3.8)	0.828
Osteoporosis	3 (12.0)	1 (7.1)	28 (13.3)	0.792
Psychiatric disorders	2 (8.0)	2 (14.3)	16 (7.6)	0.683
Positive HCV Ab, *n* (%)	4 (16.0)	4 (28.6)	33 (15.7)	0.455
HBsAg, *n* (%)	2 (8.0)	2 (14.3)	35 (16.6)	0.526
Multimorbidity, *n* (%)	8 (32.0)	6 (42.8)	81 (38.6)	0.761
Polypharmacy, *n* (%)	2 (8.0)	0 (0.0)	16 (7.6)	0.561
Weight (kg), median (IQR)	77.0 (71.0–82.0)	77.0 (74.0–86.5)	76.0 (70.0–82.0)	0.809
BMI, median (IQR)	24.7 (24.4–28.4)	25.9 (25.5–30.6)	25.1 (22.9–27.7)	0.010
BMI class, *n* (%)				<0.001
<18.5 kg	1 (4.0)	0 (0.0)	2 (0.9)	
18.5–24.9 kg	15 (60.0)	0 (0.0)	99 (47.1)	
25.0–29.9 kg	9 (36.0)	9 (64.3)	77 (36.7)	
≥30.0 kg	0 (0)	4 (28.6)	32 (15.2)	
Diet, *n* (%)				0.203
No food restrictions	21 (84.0)	10 (71.4)	182 (86.6)	
Vegan/vegetarian	1 (4.0)	0 (0.0)	7 (3.3)	
Ketogenic	1 (4.0)	2 (14.3)	10 (4.8)	
Low carbs/fats	2 (8.0)	2 (14.3)	5 (2.4)	
Others	0 (0.0)	0 (0.0)	6 (2.8)	
Physical activity, *n* (%)				0.791
None	15 (60.0)	9 (64.3)	136 (64.8)	
Occasional	2 (8.0)	1 (7.1)	7 (3.3)	
Regular	8 (32.0)	4 (28.6)	67 (31.9)	
Total cholesterol, mmol/L, median (IQR)	4.56 (4.01–5.20)	4.88 (3.93–5.32)	4.78 (4.09–5.54)	0.612
LDL, mmol/L, median (IQR)	3.01 (2.60–3.28)	3.05 (2.63–3.72)	3.11 (2.40–3.68)	0.368
HDL, mmol/L, median (IQR)	1.28 (0.99–1.64)	1.18 (1.00–1.72)	1.26 (1.04–1.47)	0.857
Triglycerides, mmol/L, median (IQR)	1.20 (0.86–1.49)	0.97 (0.75–1.89)	1.32 (0.88–1.99)	0.147
Serum creatinine, mmol/L, median (IQR)	89.0 (69.5–101.5)	86.5 (80.2–96.5)	86.0 (76.0–100.0)	0.909
eGFR, mL/min, median (IQR)	80.0 (70.5–104.5)	83.0 (79.5–97.0)	87.5 (73.8–99.0)	0.580
Reasons for switch, *n* (%)				0.188
Proactive	14 (56.0)	12 (85.7)	136 (64.8)	
Switch to STR	10 (40.0)	2 (14.3)	54 (25.7)	
DDIs	0 (0)	0 (0.0)	7 (3.3)	
Viral failure	1 (4.0)	0 (0.0)	10 (4.8)	
Drug toxicity	0 (0)	0 (0.0)	3 (1.4)	

MSM = Men who have sex with men, IVDU = intravenous drug use, IQR = interquartile range, nNRTI = non-nucleos(t)ide reverse transcriptase inhibitors, PI = protease inhibitors, INSTIs = integrase inhibitors, STR = single tablet regimen, DDIs = drug–drug interaction, BMI = body mass index.

## Data Availability

All data are herein available.

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
