# Peer review of "Viro-Immunological, Clinical Outcomes and Costs of Switching to BIC/TAF/FTC in a Cohort of People Living with HIV: A 48-Week Prospective Analysis"

_biomedicines, 2022, doi:10.3390/biomedicines10081823_

Round 1

Reviewer 1 Report

This is a timely single-centre study on individuals who switched to BIC/FTC/TAF in a single centre in Padua, Italy and who were followed for 48 weeks to analyse outcomes and discontinuations from this ART regimen. The authors show BIC/FTC/TAF is well tolerated and associated with good virological control and increases in CD4 cell count and CD4/CD8 cell ratios. They also document increases in weight and BMI, as has been shown in other contexts, though these changes did not appear to be clinically relevant. The authors also indicate that the BIC/FTC/TAF regimen may be less expensive than the individuals’ previous regimen, although a full cost effectiveness analysis was not performed. This represents an important evaluation of this modern INSTI-containing ARV regimen in a real-world context, although with small numbers of participants the study has to be mainly descriptive.

Comments:

11.     Please clarify the dates of the study: If the study enrolled anyone who switched to the BIC-containing regimen between February 1st, 2020 and December 31st, 2021 – how can individuals who started towards the end of this time window have completed their 48 weeks follow-up, and what proportion of individuals did not complete 48 week follow-up?

22.     Section 3.2 for those who discontinued BIC/FTC/TAF vs. those who did not – the numbers do not quite add up: 41 patients (14.1%) discontinued (also in Table 2), but on p. 4 discontinuation reasons are listed for 43 individuals (14.8%).

33.     Table 2 summarises and cross-tabulates various factors with the groups of individuals who discontinued their BIC-containing regimen and those who did not. Please include information on which significance tests were used in the table legend.

a.      Note this table shows associations, one variable at the time (but is not really univariate logistic regression as suggested on p.4)

b.      Table 2 does not show a significant association with Regimen before switch (as stated on P.4), with the chi2 p-value given as 0.057

c.      There is no need to highlight only those p values considered significant (e.g. by making them bold)

d.      It may be possible to combine Table 1 and 2 (show the Totals, and then also stratified by discontinued/LTFU or stayed on the drug), which would be easier to read

44.     If you have performed univariable logistic regression analysis, you could list the results in Table 3 also (i.e. show both the unadjusted and adjusted odds ratios in the table)

a.      Please clarify all the variables that have been adjusted for: Just sex and age? Were all variables adjusted in the same way?

b.      Please clarify the meaning of (n)N-2NRTIs

c.      For multivariable regression for the continuous variables (age, time since HIV diagnosis) give the units for the effect (e.g. per year older or per x years older etc)

d.      The variables age and time since HIV diagnosis are likely to be collinear, so adjusting for both in the same model may not be valid – please show the univariable results also.

55.     The discussion of increased loss to follow-up among black African individuals (p. 11) is important; since ethnicity data was collected (p.3) and used in adjustments (p.4), please list ethnicity also in Table 1 and Table 2 etc.

66.      Please include confidence intervals for the proportions shown in Figure 1 (exact methods should be used where n <20)

77.      Please use Person first language (https://peoplefirstcharter.org/ ; https://img1.wsimg.com/blobby/go/307bf032-fd32-46de-894d-184dd697d7d1/People%20first%20charter%20language%20v3%2019042022.pdf ). There are frequent references to individuals as ‘subjects’ or ‘patients’, or ‘HIV infected’: try persons, individuals, PLWH. ‘Median time of HIV infection’ could be ‘median time since diagnosis’ etc.

88.      Re writing would benefit from help from a native English speaker, and careful proof-reading

Minor corrections and grammar / language points:

-        - retro-transcriptase (abstract) – would normally be reverse transcriptase

-       -  Integrase inhibitors are more usually abbreviated as INSTIs (for integrase strand-transfer inhibitors) rather than INI, non-nucleoside RT inhibitors with a capital N (NNRTI)

-        - End of abstract - … Our results showed – should be present tense

-      -  End of abstract (p.2): week 48 equally to a mean save of 1.533 euros for year per patient: This should be: Equivalent to a mean saving of 1533 euros/year/person (in English, the . is read as a decimal point, therefore should use a comma or just give a 4-figure number. The 1.533 quantity is also mentioned on P.4 and P.12)

-      -   Introduction p. 2: … the HIV management of HIV today – one of the HIV is redundant

-      -  P.2 introduction para 2 - a spurious e

-        - The abbreviations of COPD or ITT etc are not explained

-        - P.3 cohort description: About 12 of the patients had a previous AIDS event in their past medical history – should be 12% (see table 1, 35 individuals = 12.1%)

-        - P.4 …’difficulty to swallow the table’ – should be tablet

-        - Table 1: Lenght – should be Length (also in Table 5); Ischaemic hearth disease – should be Ischaemic heart disease (also in Table 5 and text P. 3)

-        - Table 3 Polypharmacy (ref. <5 drugs/die) per day?

-        - Table 4: LDL: The units are repeated

-        - Table 5: From TA – should be TAF

-        - P.10 Discussion line 1 - one of the last STR introduced - latest?

-        - Consistency: bictegravir/emtricitabine/tenofovir alafenamide (BIC/FTC/TAF), BIC / FTC / TAF, BIC/F/TAF

-        - Ref 23 is labelled non trovo

Author Response

Reviewer #1

This is a timely single-centre study on individuals who switched to BIC/FTC/TAF in a single centre in Padua, Italy and who were followed for 48 weeks to analyze outcomes and discontinuations from this ART regimen. The authors show BIC/FTC/TAF is well tolerated and associated with good virological control and increases in CD4 cell count and CD4/CD8 cell ratios. They also document increases in weight and BMI, as has been shown in other contexts, though these changes did not appear to be clinically relevant. The authors also indicate that the BIC/FTC/TAF regimen may be less expensive than the individuals’ previous regimen, although a full cost effectiveness analysis was not performed. This represents an important evaluation of this modern INSTI-containing ARV regimen in a real-world context, although with small numbers of participants the study has to be mainly descriptive.

Dear Reviewer, we really appreciated your suggestions and comments finalized to improve the quality of our work. We reviewed extensively the text accordingly, and we hope that the manuscript is now acceptable for publication.

Comments:

  1. Please clarify the dates of the study: If the study enrolled anyone who switched to the BIC-containing regimen between February 1st, 2020 and December 31st, 2021– how can individuals who started towards the end of this time window have completed their 48 weeks follow-up, and what proportion of individuals did not complete 48 week follow-up?

You are right, we made a mistake, inclusion of patients ended up in October 2021. From October to December 2021, we collected data. This is now specified both in the abstract and in methods.

  1. Section 3.2 for those who discontinued BIC/FTC/TAF vs. those who did not – the numbers do not quit add up: 41 patients (14.1%) discontinued (also in Table 2), but on p. 4 discontinuation reasons are listed for 43 individuals (14.8%).

      We did not clearly report the data: we had 41 subjects who discontinued the drug but for two of them here were two main reasons for discontinuation (therefore it appeared 43 reasons): both the two subjects discontinued BIC/TAF/FTC due to weight gain plus intolerance (one for insomnia and mood disorders and the second for gastrointestinal complaints). We have now rephrased the sentence in page 4 to better explain the apparent contradiction in numbers (see lines 162-166).

  1. Table 2 summarizes and cross-tabulates various factors with the groups of individuals who discontinued their BIC-containing regimen and those who did not. Please include information on which significance tests were used in the table legend.

 We have now included the requested specification in Tab.2 legend.

  1. Note this table shows associations, one variable at the time (but is not really univariate logistic regression as suggested on p.4)

Apologies, we removed the part of the sentence in page 4 stating that it was a univariate analysis.

  1. Table 2 does not show a significant association with Regimen before switch (as stated on P.4), with the chi2 p-value given as 0.057

We have modified the sentence in page 4.

  1. There is no need to highlight only those p values considered significant (e.g. by making them bold)

We have now removed the bold formatting.

  1. It may be possible to combine Table 1 and 2 (show the Totals, and then also stratified by discontinued/LTFU or stayed on the drug), which would be easier to read

           We have now unified Table 1 and 2 as you requested.

  1. If you have performed univariable logistic regression analysis, you could list the results in Table 3 also (i.e. show both the unadjusted and adjusted odds ratios in the table)
  2. Please clarify all the variables that have been adjusted for: Just sex and age? Were all variables adjusted in the same way?
  3. Please clarify the meaning of (n)N-2NRTIs
  4. For multivariable regression for the continuous variables (age, time since HIV diagnosis) give the units for the effect (e.g. per year older or per x years older etc.)
  5. The variables age and time since HIV diagnosis are likely to be collinear, so adjusting for both in the same model may not be valid – please show the univariable results also.

      Thanks for your suggestion, we have now modified ex Table 3 (now Table 2), showing uni- and multivariate results. We have added a specification in the table legend reporting which variables have been included in the final multivariate model (we decided to also include age, sex, and ethnicity despite no significance in univariate, and we presented this model, but we have also run a second model including only univariate-significant variables with no changes in the results). We also included a legend for acronyms such as NNRTI etc. We specified for continuous variables the units for the effects. Compared to the old model, length of HIV infection did not associate with BIC discontinuation at univariate, and we removed from the multivariate model (also avoiding collinearity issue with age); apologies for the previous mistake due to variables selection based on comparisons results rather than univariate results. Therefore, we have also modified in the results and discussion sections any comment about the length of infection.

  1. The discussion of increased loss to follow-up among black African individuals (p. 11) is important; since ethnicity data was collected (p.3) and used in adjustments (p.4), please list ethnicity also in Table 1 and Table 2 etc.

Thanks for this observation. However, we feel that separate them will not be useful or will not add more information to the manuscript, this is why we mention just a dichotomic category (Caucasian vs. non-Caucasian). However, please find enclosed the ethnicity distribution: 54 persons of whom 45 black African, 6 Caribbean, and three from China. We mention this also in the result section.

  1. Please include confidence intervals for the proportions shown in Figure 1 (exact methods should be used where n <20)

      Thanks for this observation, Figure 1 represents punctual values that are the proportions of subjects per BMI category over the whole study population who completed the follow up (n=249), and the exact absolute number and relative % are reported in Table 4 so that there are no confidence intervals nor any type of statistical tests to be reported or applied here. We included the figure just to better represent graphically and immediately the distribution of BMI class before and after the switch.

  1. Please use Person first language (https://peoplefirstcharter.org/ ; https://img1.wsimg.com/blobby/go/307bf032-fd32-46de-894d-184dd697d7d1/People%20first%20charter%20language%20v3%2019042022.pdf ). There are frequent references to individuals as ‘subjects’ or ‘patients’, or ‘HIV infected’: try persons, individuals, PLWH. ‘Median time of HIV infection’ could be ‘median time since diagnosis’ etc.

Thanks for your suggestions, we amended the text accordingly.

  1.       Re writing would benefit from help from a native English speaker, and careful proof-reading

Minor corrections and grammar / language points:

-        - retro-transcriptase (abstract) – would normally be reverse transcriptase

-       -  Integrase inhibitors are more usually abbreviated as INSTIs (for integrase strand-transfer inhibitors) rather than INI, non-nucleoside RT inhibitors with a capital N (NNRTI)

-        - End of abstract - … Our results showed – should be present tense

-      -  End of abstract (p.2): week 48 equally to a mean save of 1.533 euros for year per patient: This should be: Equivalent to a mean saving of 1533 euros/year/person (in English, the . is read as a decimal point, therefore should use a comma or just give a 4-figure number. The 1.533 quantity is also mentioned on P.4 and P.12)

-      -   Introduction p. 2: … the HIV management of HIV today – one of the HIV is redundant

-      -  P.2 introduction para 2 - a spurious e

-        - The abbreviations of COPD or ITT etc are not explained

-        - P.3 cohort description: About 12 of the patients had a previous AIDS event in their past medical history – should be 12% (see table 1, 35 individuals = 12.1%)

-        - P.4 …’difficulty to swallow the table’ – should be tablet

-        - Table 1: Lenght – should be Length (also in Table 5); Ischaemic hearth disease – should be Ischaemic heart disease (also in Table 5 and text P. 3)

-        - Table 3 Polypharmacy (ref. <5 drugs/die) per day?

-        - Table 4: LDL: The units are repeated

-        - Table 5: From TA – should be TAF

-        - P.10 Discussion line 1 - one of the last STR introduced - latest?

-        - Consistency: bictegravir/emtricitabine/tenofovir alafenamide (BIC/FTC/TAF), BIC / FTC / TAF, BIC/F/TAF

-        - Ref 23 is labelled non trovo

Thanks for all these advises; we amended the text according to your suggestion.

Reviewer 2 Report

Major comments:

1.       In the discussion, the authors mention the presence of NRTI resistance mutations in their cohort. What proportion of patients in their cohort had these mutations, were they all in virologic failure, and did they all maintain the switched regimen? Their claim that the switched regimen overcomes resistance-induced virologic failure is not supported by the data presented here.

2.       There is a substantial overestimation of the significance of the data and a tendency toward overinterpretation. For instance, an increase of 45 CD4+ T cells/mm3 over a 48-week-period in a cohort of patients with an average CD5+ T cell count of 568 and a mostly controlled disease is not remarkable. Similarly, the type of regimen before the switch is not significantly different between patients who continued BIC and patients who switched back (p = 0.057, Table 2). Length of HIV infection is barely significant (p = 0.048). The interpretations need to be toned down to match the magnitude of changes in the data.

3.       Why do the authors think they saw a higher incidence of toxicities than other studies? They just mention various studies with lower toxicity levels and give no possible reason for the higher incidence in their cohort.

4.       In the discussion, the authors mention alcohol consumption as one of the lifestyle change parameters. There is no information on this at baseline or W48. Also, when describing dietary restrictions/changes, the authors only mention “no dietary restriction” as a recorded parameter. But in the discussion, they mention a high-fat diet. How many patients in their cohort were on a high-fat diet? Without such information, an interpretation like “the failure to change the lifestyle (reducing a high-fat diet and alcohol consumption and improving physical exercise) may have a role in the significant increase in BMI in our cohort of patients” should not be made.

5.       Similarly, how clinically significant is the average weight gain of 1 kg at W48? Also, an average BMI of 25.2 seems high; is this normal in the PLWH cohort seen at this center? How does this average BMI compare to other reported cohorts?

Minor comments:

1.       I am not sure if the journal recommends a specific length, but the abstract is too long and must be shortened. A lot of methodological details in the abstract can be removed. Also, while the authors specifically mark the “aims” section in the abstract, the rest of the abstract is unstructured.

2.       There are numerous language usage errors in the manuscript (for instance: “number of daily pill burden” should be pill burden, “accordingly to” should be “according to”). There are also many grammatical, sentence construction, and spelling mistakes throughout the text (some of them seem to be translation issues and make interpretation ambiguous). The manuscript requires substantial editing.

3.       The sentence “In this specific population, the efficacy, tolerability, and lack of resistance of B/F/TAF demonstrated in several clinical trials may represent a good treatment option.” needs citation.

4.       Table 3, why do factors like chronic renal failure and psychiatric disorders not have a reference when performing the multivariate analysis?

Author Response

Reviewer #2

  1. In the discussion, the authors mention the presence of NRTI resistance mutations in their cohort. What proportion of patients in their cohort had these mutations, were they all in virologic failure, and did they all maintain the switched regimen? Their claim that the switched regimen overcomes resistance-induced virologic failure is not supported by the data presented here.

Dear Reviewer, thanks for this observation. For the sake of your rapid assessment and to support the statement in the discussion section, we add numbers and proportions of persons with HIV who had resistance mutations in Table 1.

  1. There is a substantial overestimation of the significance of the data and a tendency toward overinterpretation. For instance, an increase of 45 CD4+ T cells/mmover a 48-week-period in a cohort of patients with an average CD5+ T cell count of 568 and a mostly controlled disease is not remarkable. Similarly, the type of regimen before the switch is not significantly different between patients who continued BIC and patients who switched back (p = 0.057, Table 2). Length of HIV infection is barely significant (p = 0.048). The interpretations need to be toned down to match the magnitude of changes in the data.

We agree with this comment, indeed we were cautious in interpreting our results and possible clinical impact of these from the beginning. However, as per you suggestion we de-emphasized the significance of such data in the discussion section.

  1. Why do the authors think they saw a higher incidence of toxicities than other studies? They just mention various studies with lower toxicity levels and give no possible reason for the higher incidence in their cohort.

Thanks for this observation. Even if we not specifically assessed this point, we feel that COVID-19 pandemic may have had an impact, by producing stress and anxiety that could have accentuated or triggered the perception of some side effects. We added a brief paragraph to explain and discuss this into the discussion section.

  1. In the discussion, the authors mention alcohol consumption as one of the lifestyles change parameters. There is no information on this at baseline or W48. Also, when describing dietary restrictions/changes, the authors only mention “no dietary restriction” as a recorded parameter. But in the discussion, they mention a high-fat diet. How many patients in their cohort were on a high-fat diet? Without such information, an interpretation like “the failure to change the lifestyle (reducing a high-fat diet and alcohol consumption and improving physical exercise) may have a role in the significant increase in BMI in our cohort of patients” should not be made.

Dear reviewer, you are right, since we did not mention/collect data about alcohol consumption or high-fat diet, we rephrased the sentences into the discussion section.

  1. Similarly, how clinically significant is the average weight gain of 1 kg at W48? Also, an average BMI of 25.2 seems high; is this normal in the PLWH cohort seen at this center? How does this average BMI compare to other reported cohorts?

Thanks for this observation that gave us the opportunity to compare ours with other cohorts, and to extensively improve discussion on this purpose (please see line 304-325).

Minor comments:

  1. I am not sure if the journal recommends a specific length, but the abstract is too long and must be shortened. A lot of methodological details in the abstract can be removed. Also, while the authors specifically mark the “aims” section in the abstract, the rest of the abstract is unstructured.

Abstract was shortened and structured.

  1. There are numerous language usage errors in the manuscript (for instance: “number of daily pill burden” should be pill burden, “accordingly to” should be “according to”). There are also many grammatical, sentence construction, and spelling mistakes throughout the text (some of them seem to be translation issues and make interpretation ambiguous). The manuscript requires substantial editing.

DONE

  1. The sentence “In this specific population, the efficacy, tolerability, and lack of resistance of B/F/TAF demonstrated in several clinical trials may represent a good treatment option.” needs citation.

DONE

  1. Table 3, why do factors like chronic renal failure and psychiatric disorders not have a reference when performing the multivariate analysis?+

Amended.

Round 2

Reviewer 2 Report

Comments:

1.       Grammar is still an issue. The introduction section of the newly restructured abstract, for instance, is a run-on sentence that needs to be rewritten. More such mistakes are present throughout the text.

2.       Resistance mutation testing information should be included in the methods section. Also, include if all the patients were assessed and the efficiency of the tests.

3.       A few citations for the authors’ discussion on the role COVID lockdowns may have had on PLWH cohorts will be helpful to bolster their speculation. Such studies were done and are available online.

4.       Do the authors think a longer follow-up period may also have been useful in getting more differences from baseline after the switch? They should add that to the discussion section.

5.       Line 178, a p = 0.057 is not a mild trend towards significance; it is only a mild trend. Please reword.

Author Response

Reviewer #2

Comments:

  1. Grammar is still an issue. The introduction section of the newly restructured abstract, for instance, is a run-on sentence that needs to be rewritten. More such mistakes are present throughout the text.

Dear Reviewer, thank you. The whole text has been revised to amend any possible mistakes.

  1. Resistance mutation testing information should be included in the methods section. Also, include if all the patients were assessed and the efficiency of the tests.

Dear Reviewer, information about the resistance mutation testing information, as well as the method with which they were obtained was added into methods. In our center, anytime a patient experiences a virological failure, resistance test by Sanger sequencing methods on plasma samples is performed. Even if this test only detects viral variants with a prevalence higher than 20%, is the most common approach in clinical practice (Larder BA et al, Nature 1993, Palmer S, et al. J Clin Microbiol 2005, Alidjinou EK et al. L Antimicrob Chemother, 2017). Unfortunately, 43/290 (14.8%) of our patients in our cohort did not have a genotypic resistance test since they were diagnosed with HIV from 1985 to 1996. We added a brief sentence also for this.

  1. A few citations for the authors’ discussion on the role COVID lockdowns may have had on PLWH cohorts will be helpful to bolster their speculation. Such studies were done and are available online.

Thanks for this observation, some references on this purpose have been added, as per your suggestions.

  1. Do the authors think a longer follow-up period may also have been useful in getting more differences from baseline after the switch? They should add that to the discussion section.

Dear Reviewer, thanks for this observation. It is difficult to speculate on a possible evolution of all the different parameters over time. However, we feel that weigh will continue to increase (from of a mean kg/year as already shown in the general population, to 5-6 kilograms during 5 years of follow-up as already observed in clinical trials on integrase inhibitors combined with tenofovir alafenamide). We added a brief comment on this in the discussion section.

  1. Line 178, a p = 0.057 is not a mild trend towards significance; it is only a mild trend. Please reword.

Amendend.